A new species of lotic breeding salamander (Amphibia, Caudata, Hynobiidae) from Shikoku, Japan

Kanamori Sally 1 s.kanamori.5328@gmail.com
http://orcid.org/0000-0002-6274-4959 Nishikawa Kanto 1 2
Matsui Masafumi 2
Tanabe Shingo 1
1 Graduate School of Global and Environmental Studies, Kyoto University , Kyoto , Japan
2 Graduate School of Human and Environmental Studies, Kyoto University , Kyoto , Japan
Iqbal Muhammad Aamir
Electronic publication date: 2022 Aug 26
Publication date: 2022
Volume: 10
Electronic Location ID: e13891
Received 2022 May 2; Accepted 2022 Jul 21
Copyright: © 2022 Kanamori et al.
Copyright year: 2022
Copyright holder: Kanamori et al.
License: This is an open access article distributed under the terms of the Creative Commons Attribution License, which permits unrestricted use, distribution, reproduction and adaptation in any medium and for any purpose provided that it is properly attributed. For attribution, the original author(s), title, publication source (PeerJ) and either DOI or URL of the article must be cited.
License URL: https://creativecommons.org/licenses/by/4.0/

Keywords: Hynobius, Single nucleotide polymorphism, Allopatric speciation, Introgression

Funding: Environment Research and Technology Development Fund JPMEERF20204002 JST SPRING JPMJSP2110 This work was supported by the Environment Research and Technology Development Fund (JPMEERF20204002) of the Environmental Restoration and Conservation Agency of Japan and the JST SPRING, Grant Number JPMJSP2110. There was no additional external funding received for this study. The funders had no role in study design, data collection and analysis, decision to publish, or preparation of the manuscript.

==============================
Background

Hynobius hirosei is a lotic-breeding salamander endemic to Shikoku Island in western Japan. Significant allozymic and morphological differences have been found among the populations of this species; however, the degree and pattern of intraspecific variation have not been surveyed using a sufficient number of samples.

Methods

For the taxonomic revision of H. hirosei, we conducted genetic and morphological surveys using samples collected throughout the distribution. Phylogenetic analysis using the cytochrome b region of mitochondrial DNA and population structure analysis using single nucleotide polymorphisms were conducted to evaluate the population structure within the species and the degree of genetic differentiation. Subsequently, a morphological survey based on multivariate and univariate analyses was performed to assess the morphological variation.

Results

Genetic analyses revealed three genetic groups (Tsurugi, Central, and Nanyo) within H. hirosei, with the Nanyo group distributed allopatrically from the others, and the Tsurugi and Central groups distributed parapatrically with the formation of a hybrid zone between them. The Nanyo group was morphologically distinguishable from the remaining samples, including the topotype of H. hirosei, based on a smaller body size and several ratio values of characters to snout-vent length, longer axilla-groin distance, shorter tail length, shorter internarial distance, longer upper eyelid length, and larger medial tail width. These results support the notion that the Nanyo group is an undescribed species. However, the remaining genetically differentiated groups could not be divided in the present study. Herein, we described the Nanyo group as a new species.

Introduction

Hynobius Tschudi, 1838 is a salamander distributed in Eastern and Central Asia (Frost, 2022). Hynobius usually resides in water during the larval period, inhabits the forest floor after metamorphosis, and gathers in the water body to breed. Each species has a different habit and belongs to three breeding types: lentic breeding, which involves the deposit of eggs in pools, sulculus, and ponds; lotic breeding, which involves the deposit of eggs in streams; and underground breeding, which involves the deposit of eggs in underground water (Herpetological Society of Japan, 2021). The Japanese archipelago is known as a center of diversity of the genus, as 39 of 58 species (67.2%) of Hynobius are endemic to Japan (Frost, 2022). The number of species of Japanese Hynobius has markedly increased over the last three decades owing to progress in taxonomic studies using molecular approaches to assess allozymic data, mitochondrial DNA (mtDNA), and microsatellite markers (Matsui & Miyazaki, 1984; Matsui, 1987; Matsui et al., 2004, 2017; Nishikawa & Matsui, 2014; Okamiya et al., 2018; Sugawara et al., 2018, 2021, 2022a, 2022b; Tominaga, Matsui & Nishikawa, 2019a, 2019b). However, some species complexes remain taxonomically unresolved owning to the paucity of genome-wide information. Recently, genome-wide analyses, such as population structure analysis using single nucleotide polymorphisms (SNP), have contributed to delimiting species and/or detecting hybridization among recognized genetic groups (Colliard et al., 2010; Kindler et al., 2017). Using genome-wide SNP generated by multiplexed ISSR genotyping by sequencing (MIG-seq), Matsui et al. (2019) comprehensively investigated the genetic identities of the Japanese clouded salamander, H. nebulosus, and its closest relatives and then detected eight cryptic species in H. nebulosus. MtDNA remains a useful tool for the initial detection of cryptic lineages owing to their small effective population size, low recombination rate, and high substitution rate (Burbrink & Ruane, 2021).

Hynobius hirosei Lantz, 1931 is a lotic breeding type that is known to widely occur in Shikoku (Sato, 1934; Nishikage, 1960; Tanabe & Okayama, 1990; Kawata, 1992; Okayama, 1995a, 2004; Tanabe & Matsui, 2001; Uwa, Tanabe & Okayama, 2004; Tamura, 2012; Kaneshiro et al., 2021). The habitats of the species are high elevation areas ranging from 650 to 1,700 m (Okayama, 1995b). The vegetation in these habitats is deciduous broad-leaved forest, artificial forest, and mixed forest, but is often deciduous broad-leaved forest (Okayama, 1995b; Watanabe et al., 2015). Hynobius hirosei sympatrically occurs with Onychodactylus kinneburi and H. kuishiensis (or H. tsurugiensis; Tominaga et al., 2019b). In fact, these species reside in separate habitats during spawning and larval periods (Uwa, 1979), but share terrestrial habitats after metamorphosis and out of the breeding season. Hynobius hirosei was described based on two females collected from Mt. Ishizuchi, Ehime Prefecture, but was synonymized with Pachypalaminus boulengeri (H. boulengeri, presently) from the Kii Peninsula of mainland Honshu owning to its morphological similarity (Sato, 1934). By examining allozymic variation in H. boulengeri, Nishikawa et al. (2001) found that the species was separated at distinct specific levels into three groups: the Kii Peninsula on the Honshu mainland, Shikoku Island, and Kyushu Island groups. Thereafter, Nishikawa et al. (2007) revived the name, H. hirosei, for the Shikoku population based on morphological and allozymic differences, and revealed large intraspecific variation within H. hirosei. However, no studies have investigated the population structure and intraspecific variation of H. hirosei using comprehensive data collected from its distribution, despite the possible inclusion of undescribed species within the salamander.

In this study, to determine or assign the taxonomic status of divergent populations within H. hirosei, we conducted phylogenetic analysis using mtDNA, genetic structure analysis using SNP, and morphological analyses, which revealed fine-scale population structure and intraspecific variation.

Materials and Methods

Nomenclatural acts

The electronic version of this article in portable document format will represent a published work according to the International Commission on Zoological Nomenclature (ICZN), and hence the new names contained in the electronic version are effectively published under that Code from the electronic edition alone (see Articles 8.5–8.6 of the Code). This published work and the nomenclatural acts it contains have been registered in ZooBank, the online registration system for the ICZN. The ZooBank Life Science Identifiers (LSIDs) can be resolved and the associated information can be viewed through any standard web browser by appending the LSID to the prefix http://zoobank.org/. The LSID for this publication is as follows: urn:lsid:zoobank.org:pub: 1EFFBF1B-1249-4A9F-8AB2-36BFCD6F6B9E. The online version of this work is archived and available from the following digital repositories: PeerJ, PubMed Central and CLOCKSS.

Sample collection

Using the protocol of Nishikawa et al. (2007), we collected salamanders in the field and fully anesthetized them with an acetone-chloroform solution for subsequent processes. Liver and muscle tissues were collected from the anesthetized salamanders and preserved in 99% ethanol or stored in a deep-freezer for molecular analyses. Voucher specimens were then fixed in 10% formalin, and later preserved in 70% ethanol for permanent storage at the Graduate School of Human and Environment Studies, Kyoto University (KUHE). Animal collection and experiments followed the guideline of animal experiments of the university and were approved by the animal experimentation ethics committee at the KUHE (certificate number: 29–A–7, 30–A–7, 20–A–7, 20–A–5).

Genetic sample and DNA extraction

To survey intraspecific genetic variation in H. hirosei, 116 samples covering the entire geographic range of Shikoku were employed (Fig. 1A, Table 1). Total genomic DNA was extracted from the tissues using the DNeasy Blood & Tissue Kit (Qiagen, Hilden, Germany).

Figure 1 (A) Sampling localities of Hynobius hirosei sensu lato. (B) Maximum likelihood (ML) tree based on the complete cyt b gene for H. hirosei sensu lato and the other hynobiid species.

(A) Blue triangles: the Tsurugi group; orange circles: the Central group; orange star: type locality of H. hirosei; green diamonds: the Nanyo group (H. oni); green star: type locality of H. oni. Major rivers are depicted in blue. The map was generated and modified from https://www.gsi.go.jp/LAW/2930/index.html (in Japanese). For locality numbers, refer to Table 1. (B) Closed black circles indicate significant supports by BS > 70 for ML. Numbers preceded by “L” indicate locality number.

Table 1 Information of samples used for the genetic analyses in this study. Locality numbers correspond to Fig. 1A.

Species	Clade	Locality no.	Locality	Voucher	Genbank accession no. (mtDNA)	DDBJ accession no. (SNP)	Reference	
Hynobius hirosei	Tsurugi group	1	Sanagochi Village, Tokushima Prefecture	KUHE T2971	ON110825	DRR361514	This study	
	Tsurugi group	2	Naka Town, Tokushima Prefecture	KUHE 61326	ON110826	DRR361515	This study	
	Tsurugi group	2	Naka Town, Tokushima Prefecture	KUHE 61327	ON110827	DRR361516	This study	
	Tsurugi group	2	Naka Town, Tokushima Prefecture	KUHE unnumbered 1	ON110828	DRR361517	This study	
	Tsurugi group	2	Naka Town, Tokushima Prefecture	KUHE unnumbered 2	ON110829	DRR361518	This study	
	Tsurugi group	2	Naka Town, Tokushima Prefecture	KUHE unnumbered 3	ON110830	DRR361519	This study	
	Tsurugi group	2	Naka Town, Tokushima Prefecture	KUHE unnumbered 22	–	DRR361520	This study	
	Tsurugi group	3	Yoshinogawa City, Tokushima Prefecture	KUHE unnumbered 4	ON110831	DRR361521	This study	
	Tsurugi group	3	Yoshinogawa City, Tokushima Prefecture	KUHE unnumbered 5	ON110832	DRR361524	This study	
	Tsurugi group	3	Yoshinogawa City, Tokushima Prefecture	KUHE unnumbered 6	ON110833	DRR361522	This study	
	Tsurugi group	3	Yoshinogawa City, Tokushima Prefecture	KUHE unnumbered 7	ON110834	DRR361523	This study	
	Tsurugi group	4	Mima City, Tokushima Prefecture	KUHE T3063	ON110835	DRR361527	This study	
	Tsurugi group	4	Miyoshi City, Tokushima Prefecture	KUHE T3207	–	DRR361528	This study	
	Tsurugi group	4	Miyoshi City, Tokushima Prefecture	KUHE T3470	ON110836	–	This study	
	Tsurugi group	4	Miyoshi City, Tokushima Prefecture	KUHE T3497	ON110837	DRR361529	This study	
	Tsurugi group	4	Miyoshi City, Tokushima Prefecture	KUHE T3802	ON110838	DRR361530	This study	
	Tsurugi group	4	Miyoshi City, Tokushima Prefecture	KUHE T3803	ON110839	DRR361531	This study	
	Tsurugi group	4	Miyoshi City, Tokushima Prefecture	KUHE unnumbered 8	ON110840	DRR361525	This study	
	Tsurugi group	4	Miyoshi City, Tokushima Prefecture	KUHE unnumbered 9	ON110841	DRR361526	This study	
	Tsurugi group	5	Mima City, Tokushima Prefecture	KUHE T3449	ON110842	DRR361532	This study	
	Tsurugi group	6	Umaji Village, Kochi Prefecture	KUHE unnumbered 10	ON110843	DRR361533	This study	
	Central group	7	Miyoshi City, Tokushima Prefecture	KUHE 62193-1	ON110855	DRR361536	This study	
	Central group	7	Miyoshi City, Tokushima Prefecture	KUHE 62193-2	ON110856	DRR361537	This study	
	Central group	7	Miyoshi City, Tokushima Prefecture	KUHE 62193-3	ON110857	DRR361538	This study	
	Central group	7	Miyoshi City, Tokushima Prefecture	KUHE 62193-4	ON110858	DRR361539	This study	
	Central group	7	Miyoshi City, Tokushima Prefecture	KUHE 62193-5	ON110859	DRR361540	This study	
	Central group	7	Miyoshi City, Tokushima Prefecture	KUHE 62193-6	ON110860	DRR361541	This study	
	Central group	7	Miyoshi City, Tokushima Prefecture	KUHE 62193-7	ON110861	DRR361542	This study	
	Central group	7	Miyoshi City, Tokushima Prefecture	KUHE 62193-8	ON110862	DRR361543	This study	
	Central group	7	Miyoshi City, Tokushima Prefecture	KUHE 62195	ON110863	DRR361534	This study	
	Central group	7	Miyoshi City, Tokushima Prefecture	KUHE 62237	ON110864	DRR361535	This study	
	Central group	8	Mannou Town, Kagawa Prefecture	KUHE 9677	ON110844	DRR361544	This study	
	Central group	8	Mannou Town, Kagawa Prefecture	KUHE 9678	ON110845	DRR361545	This study	
	Central group	8	Mannou Town, Kagawa Prefecture	KUHE 9679	ON110846	DRR361546	This study	
	Central group	8	Mannou Town, Kagawa Prefecture	KUHE 9680	ON110847	DRR361547	This study	
	Central group	8	Mannou Town, Kagawa Prefecture	KUHE 9681	ON110848	DRR361548	This study	
	Central group	8	Mannou Town, Kagawa Prefecture	KUHE 9682	ON110849	DRR361549	This study	
	Central group	8	Mannou Town, Kagawa Prefecture	KUHE 9683	ON110850	DRR361550	This study	
	Central group	8	Mannou Town, Kagawa Prefecture	KUHE 9684	ON110851	DRR361551	This study	
	Central group	8	Mannou Town, Kagawa Prefecture	KUHE unnumbered 11	ON110852	DRR361552	This study	
	Central group	8	Mannou Town, Kagawa Prefecture	KUHE unnumbered 12	ON110853	DRR361553	This study	
	Central group	9	Kannonji City, Kagawa Prefecture	KUHE T3269	ON110854	DRR361554	This study	
	Central group	10	Otoyo Town, Kochi Prefecture	KUHE 62192-1	ON110876	DRR361555	This study	
	Tsurugi group	10	Otoyo Town, Kochi Prefecture	KUHE 62192-2	ON110877	DRR361556	This study	
	Tsurugi group	10	Otoyo Town, Kochi Prefecture	KUHE 62192-3	ON110878	DRR361557	This study	
	Tsurugi group	10	Otoyo Town, Kochi Prefecture	KUHE 62192-4	ON110879	DRR361558	This study	
	Tsurugi group	10	Otoyo Town, Kochi Prefecture	KUHE 62192-5	ON110880	DRR361559	This study	
	Tsurugi group	10	Otoyo Town, Kochi Prefecture	KUHE 62192-6	ON110881	DRR361560	This study	
	Tsurugi group	10	Otoyo Town, Kochi Prefecture	KUHE 62192-7	ON110882	DRR361561	This study	
	Tsurugi group	10	Otoyo Town, Kochi Prefecture	KUHE 62192-8	ON110883	DRR361562	This study	
	Central group	11	Motoyama Town, Kochi Prefecture	KUHE 24915	ON110884	DRR361563	This study	
	Central group	11	Motoyama Town, Kochi Prefecture	KUHE 62198-4	ON110885	DRR361567	This study	
	Central group	11	Motoyama Town, Kochi Prefecture	KUHE 62198-5	ON110886	DRR361568	This study	
	Central group	11	Motoyama Town, Kochi Prefecture	KUHE 62198-6	ON110887	DRR361569	This study	
	Central group	11	Motoyama Town, Kochi Prefecture	KUHE 62198-7	ON110888	DRR361570	This study	
	Central group	11	Motoyama Town, Kochi Prefecture	KUHE 62198-8	ON110889	DRR361571	This study	
	Central group	11	Motoyama Town, Kochi Prefecture	KUHE 62198-9	ON110890	DRR361572	This study	
	Central group	11	Motoyama Town, Kochi Prefecture	KUHE 62198-10	ON110891	DRR361564	This study	
	Central group	11	Motoyama Town, Kochi Prefecture	KUHE 62198-11	ON110892	DRR361565	This study	
	Central group	11	Motoyama Town, Kochi Prefecture	KUHE 62198-12	ON110893	DRR361566	This study	
	Central group	11	Motoyama Town, Kochi Prefecture	KUHE 62198-13	ON110894	–	This study	
	Central group	12	Kochi City, Kochi Prefecture	KUHE 9512	ON110895	DRR361573	This study	
	Central group	12	Kochi City, Kochi Prefecture	KUHE 9513	ON110896	DRR361574	This study	
	Central group	12	Kochi City, Kochi Prefecture	KUHE 9514	ON110897	DRR361575	This study	
	Central group	12	Kochi City, Kochi Prefecture	KUHE 9515	ON110898	DRR361576	This study	
	Central group	12	Kochi City, Kochi Prefecture	KUHE 24196	ON110899	DRR361577	This study	
	Central group	12	Kochi City, Kochi Prefecture	KUHE 24199	ON110900	DRR361578	This study	
	Central group	13	Miyoshi City, Tokushima Prefecture	KUHE 61300-1	ON110865	DRR361579	This study	
	Central group	13	Miyoshi City, Tokushima Prefecture	KUHE 61300-2	ON110866	DRR361580	This study	
	Central group	13	Miyoshi City, Tokushima Prefecture	KUHE 61300-3	ON110867	DRR361581	This study	
	Central group	14	Otoyo Town, Kochi Prefecture	KUHE 62196-1	ON110868	DRR361585	This study	
	Central group	14	Otoyo Town, Kochi Prefecture	KUHE 62196-2	ON110869	DRR361586	This study	
	Central group	14	Otoyo Town, Kochi Prefecture	KUHE 62196-3	ON110870	DRR361587	This study	
	Central group	14	Otoyo Town, Kochi Prefecture	KUHE 62196-4	ON110871	DRR361588	This study	
	Central group	14	Otoyo Town, Kochi Prefecture	KUHE 62196-5	ON110872	DRR361589	This study	
	Central group	14	Otoyo Town, Kochi Prefecture	KUHE 62196-6	ON110873	DRR361590	This study	
	Central group	14	Otoyo Town, Kochi Prefecture	KUHE 62196-7	ON110874	DRR361591	This study	
	Central group	14	Otoyo Town, Kochi Prefecture	KUHE 62197	ON110875	DRR361582	This study	
	Central group	14	Otoyo Town, Kochi Prefecture	KUHE 62235	–	DRR361583	This study	
	Central group	14	Otoyo Town, Kochi Prefecture	KUHE 62236	–	DRR361584	This study	
	Central group	15	Ino Town, Kochi Prefecture	KUHE T2161	ON110901	DRR361592	This study	
	Central group	15	Ino Town, Kochi Prefecture	KUHE T2162	ON110902	DRR361593	This study	
	Central group	15	Ino Town, Kochi Prefecture	KUHE T2163	ON110903	DRR361594	This study	
	Central group	16	Saijo City, Ehime Prefecture	KUHE unnumbered 13	ON110904	DRR361595	This study	
	Central group	16	Saijo City, Ehime Prefecture	KUHE unnumbered 14	ON110905	DRR361596	This study	
	Central group	16	Saijo City, Ehime Prefecture	KUHE unnumbered 15	ON110906	DRR361597	This study	
	Central group	17	Saijo City, Ehime Prefecture	KUHE T2173	ON110907	DRR361598	This study	
	Central group	17	Saijo City, Ehime Prefecture	KUHE T2174	ON110908	DRR361599	This study	
	Central group	17	Saijo City, Ehime Prefecture	KUHE T2175	ON110909	DRR361600	This study	
	Central group	18	Kumakogen Town, Ehime Prefecture	KUHE 24467	ON110910	DRR361601	This study	
	Central group	18	Kumakogen Town, Ehime Prefecture	KUHE 24468	ON110911	DRR361602	This study	
	Central group	18	Kumakogen Town, Ehime Prefecture	KUHE T3009	ON110912	–	This study	
	Central group	18	Kumakogen Town, Ehime Prefecture	KUHE unnumbered 16	ON110913	DRR361603	This study	
	Central group	18	Kumakogen Town, Ehime Prefecture	KUHE unnumbered 17	ON110914	DRR361604	This study	
	Central group	18	Kumakogen Town, Ehime Prefecture	KUHE unnumbered 18	ON110915	DRR361605	This study	
	Central group	19	Toon City, Ehime Prefecture	KUHE unnumbered 19	ON110916	DRR361606	This study	
	Central group	19	Toon City, Ehime Prefecture	KUHE unnumbered 20	ON110917	DRR361608	This study	
	Central group	19	Toon City, Ehime Prefecture	KUHE unnumbered 21	ON110918	DRR361607	This study	
	Central group	20	Tsuno Town, Kochi Prefecture	KUHE 28086	ON110919	DRR361609	This study	
	Central group	20	Tsuno Town, Kochi Prefecture	KUHE T2973	ON110920	DRR361610	This study	
	Central group	20	Tsuno Town, Kochi Prefecture	KUHE T2974	ON110921	DRR361611	This study	
	Central group	21	Kumakogen Town, Ehime Prefecture	KUHE 57485-1	ON110922	DRR361612	This study	
	Central group	21	Kumakogen Town, Ehime Prefecture	KUHE 57485-2	ON110923	DRR361613	This study	
	Central group	21	Kumakogen Town, Ehime Prefecture	KUHE 57485-3	ON110924	DRR361614	This study	
	Central group	22	Uchiko Town, Ehime Prefecture	KUHE T2916	ON110928	DRR361618	This study	
	Central group	23	Seiyo City, Ehime Prefecture	KUHE 57481	ON110925	DRR361616	This study	
	Central group	23	Seiyo City, Ehime Prefecture	KUHE 57482	ON110926	DRR361617	This study	
	Central group	23	Seiyo City, Ehime Prefecture	KUHE T2702	ON110927	DRR361615	This study	
Hynobius oni	Nanyo group	24	Uwajima City, Ehime Prefecture	KUHE 61097	ON110929	DRR361619	This study	
	Nanyo group	24	Uwajima City, Ehime Prefecture	KUHE 61098	ON110930	DRR361620	This study	
	Nanyo group	24	Uwajima City, Ehime Prefecture	KUHE 62785	ON110931	–	This study	
	Nanyo group	24	Uwajima City, Ehime Prefecture	KUHE 62789	ON110932	–	This study	
	Nanyo group	25	Ainan Town, Ehime Prefecture	KUHE 24085	ON110933	–	This study	
	Nanyo group	25	Ainan Town, Ehime Prefecture	KUHE 24086	ON110934	DRR361621	This study	
	Nanyo group	25	Ainan Town, Ehime Prefecture	KUHE 24087	ON110935	DRR361622	This study	
	Nanyo group	26	Ainan Town, Ehime Prefecture	KUHE 24088	ON110936	DRR361623	This study	
	Nanyo group	26	Ainan Town, Ehime Prefecture	KUHE 24800	ON110937	DRR361624	This study	
	Nanyo group	26	Ainan Town, Ehime Prefecture	KUHE T3788	ON110938	DRR361627	This study	
	Nanyo group	26	Ainan Town, Ehime Prefecture	KUHE 60947	ON110939	DRR361625	This study	
	Nanyo group	26	Ainan Town, Ehime Prefecture	KUHE 61096	ON110940	DRR361626	This study	
Hynobius sematonotos	–	–	Higashihiroshima City, Hiroshima Prefecture	KUHE 34694	LC433768	–	Tominaga, Matsui & Nishikawa (2019a)	
Hynobius katoi	–	–	Hamamatsu City, Shizuoka Prefecture	KUHE 37128	AB266673	–	Matsui et al. (2007)	
Hynobius oyamai	–	–	Kokonoe Town, Oita Prefecture	KUHE 32584	LC433769	–	Tominaga, Matsui & Nishikawa (2019a)	
Hynobius stejnegeri	–	–	Yamato Town, Kumamoto Prefecture	KUHE 28011	AB921166	–	Nishikawa & Matsui (2014)	
Hynobius naevius	–	–	Sasebo City, Nagasaki Prefecture	KUHE 28584	AB266672	–	Matsui et al. (2007)	
Hynobius ikioi	–	–	Gokase Town, Miyazaki Prefecture	KUHE 22817	AB921162	–	Nishikawa & Matsui (2014)	
Hynobius amakusaensis	–	–	Amakusa City, Kumamoto Prefecture	KUHE 30338	AB921167	–	Nishikawa & Matsui (2014)	
Hynobius osumiensis	–	–	Kinko Town, Kagoshima Prefecture	KUHE 24797	AB921165	–	Nishikawa & Matsui (2014)	
Hynobius shinichisatoi	–	–	Saiki City, Oita Prefecture	KUHE 22889	AB921163	–	Nishikawa & Matsui (2014)	
Hynobius kimurae	–	–	Otsu City, Shiga Prefecture	KUHE 16689	AB266674	–	Matsui et al. (2007)	
Hynobius boulengeri	–	–	Kamikitayama Village, Nara Prefecture	KUHE 25653	AB266675	–	Matsui et al. (2007)	
Hynobius retardatus	–	–	Ebetsu City, Hokkaido Prefecture	KUHE 13034	AB363609	–	Matsui et al. (2007)	
Salamandrella keyserlingii	–	–	Kushiro City, Hokkaido Prefecture	KUHE 13055	AB363573	–	Matsui et al. (2007)	

Mitochondrial DNA analysis

The sequence data of full-length cytochrome b (cyt b; 1,141 bp) of mtDNA were obtained using the primers HYD_Cytb_F1 5′–CYAAYCCTAAAGCWGCAAAATA–3′ (Matsui et al., 2008); salamander_cytb_R_N2 5′–YTYTCAATCTTKGGYTTACAAGACC–3′ (Matsui et al., 2008); cytb_R1_cynops 5′–AARTAYGGGTGRAADGRRAYTTTRTCT–3′ (Aoki, Matsui & Nishikawa, 2013); and cytb_F2_cynops 5′– CAYTTYYTGYTMCCATTYYTAATTGCAGG–3′ (Aoki, Matsui & Nishikawa, 2013).

The detailed experimental protocols were described by Aoki, Matsui & Nishikawa (2013). Sequences were assembled and checked using the Chromas Pro 1.34 software (Technelysium Pty Ltd., South Brisbane, QLD, Australia). The resulting sequence was deposited in GenBank (accession numbers ON110825–ON110940). For comparisons, the GenBank data of H. amakusaensis from Kumamoto Prefecture (voucher number: KUHE 30338; GenBank accession number: AB921167), H. boulengeri from Nara Prefecture (KUHE 25653: AB266675), H. ikioi from Miyazaki Prefecture (KUHE 22817: AB921162), H. katoi from Shizuoka Prefecture (KUHE 37128: AB266673), H. kimurae from Shiga Prefecture (KUHE 16689: AB266674), H. naevius from Saga Prefecture (KUHE 28584: AB266672), H. osumiensis from Kagoshima Prefecture (KUHE 24797: AB921165), H. oyamai from Oita Prefecture (KUHE 32584: LC433769), H. retardatus from Hokkaido Prefecture (KUHE 13057: AB363573), H. sematonotos from Hiroshima Prefecture (KUHE 34694; LC433768), H. shinichisatoi from Oita Prefecture (KUHE 22889: AB921163), H. stejnegeri from Kumamoto Prefecture (KUHE 28011: AB921166) and Salamandrella keyserlingii from Hokkaido Prefecture (KUHE 13057: AB363573) were added. Table 1 lists the citations that contain these sequences.

The obtained sequence data were aligned using MAFFT 7 (Katoh & Standley, 2013). Before phylogenetic analysis, we selected the best substitution model using Kakusan4 (Tanabe, 2011) based on the Akaike information criterion (Akaike, 1974) for Likelihood (ML) method and Bayesian information criterion (Schwarz, 1978) for Bayesian inference (BI) method. The ML tree was estimated using RAxML 8.2 software (Stamatakis, 2014). For the confidence of the ML tree, bootstrap analysis was conducted with 1,000 replicates. Nodes with bootstrap values (BS) >70% were considered as sufficiently supported (Huelsenbeck & Hillis, 1993). The BI tree was generated using MrBayes 3.2.6 software (Ronquist et al., 2012). The BI analyses were performed using two parallel runs of four Markov chain Monte Carlo (MCMC) methods for 20 million generations. The first 25% of the generations were discarded as a burn-in, and one of every 100 of the remaining generations was sampled. The convergence of the MCMC runs was checked using TRACER 1.6 software (Rambaut et al., 2014). The uncorrected p-distance among samples was calculated using MEGA 7 software (Kumar, Stecher & Tamura, 2016).

SNP analysis

To estimate the genetic structure of H. hirosei (114 individuals from 26 populations; DDBJ accession number: DRR361514–DRR361627), we used genome-wide SNP obtained based on MIG-seq (Suyama & Matsuki, 2015) using the Illumina MiSeq system. We followed the protocol described by Suyama & Matsuki (2015), except that an annealing temperature of 38 °C (originally, 48 °C) for the first PCR and indexed forward and reverse primers for the second PCR (originally, common forward and indexed reverse primers) were employed. The sequenced data were filtered using fastp ver. 3 (Chen et al., 2018) to trim the primer regions and remove low-quality and short reads. The qualified quality phred, length required, and front/tail trimming settings were 30, 80, and 14, respectively. After filtered reads 1 and 2 were connected to one, Stacks ver. 2.3c (Catchen et al., 2011) was executed for SNP detection using the ‘denovo_map.pl’ program as follows: number of mismatches allowed between stacks within individuals (M) = 2; and number of mismatches allowed between stacks between individuals (n) = 2. Of the several stages in the program, ‘ustacks’ was configured as follows: minimum depth of coverage required to create a stack (m) = 3 and maximum distance allowed to align secondary reads to primary stacks (N) = 2. Finally, the ‘populations’ program was set as follows: minimum percentage of samples in a population (r) = 0.7; minimum number of populations in a locus (p) = 2; and specify a minimum minor allele frequency required to process a SNP (min-maf) = 0.05; and specify a maximum observed heterozygosity required to process a SNP (max-obs-het) = 0.75.

The genetic structures within H. hirosei were estimated using Bayesian clustering in STRUCTURE ver. 2. 3. 4 (Pritchard, Stephens & Donnelly, 2000), as well as EasyParallel (Zhao et al., 2020), which is multithreaded parallelization to facilitate population structure analysis. The Monte Carlo Markov chains ran for 1 million generations, including a burn-in of 100,000 generations. This calculation was repeated ten times for the number of each cluster to evaluate genetic differentiation in nuclear markers among the clusters. Multiple STRUCTURE runs were combined using CLUMPP (Jakobsson & Rosenberg, 2007).

Morphological data and analysis

Measurements to 0.1 mm were recorded using a Mitutoyo digital caliper and a stereoscopic microscope when necessary. For the morphological comparisons, 83 adult specimens of H. hirosei were used and, following 26 metric characters and five meristic characters, were employed by referring to Nishikawa et al. (2007): 1) SVL (snout-vent length); 2) HL (head length); 3) HW (head width); 4) MXHW (maximum head width); 5) SL (snout length); 6) LJL (lower jaw length); 7) IND (internarial distance); 8) IOD (interorbital distance); 9) UEW (upper eyelid width); 10) UEL (upper eyelid length); 11) AGD (axilla-groin distance); 12) TRL (trunk length); 13) TAL (tail length); 14) BTAW (basal tail width); 15) MTAW (medial tail width); 16) BTAH (basal tail height); 17) MXTAH (maximum tail height); 18) MTAH (medial tail height); 19) FLL (forelimb length); 20) HLL (hindlimb length); 21); 2FL (second finger length); 22) 3FL (third finger length); 23) 3TL (third toe length); 24) 5TL (fifth toe length); 25) VTW (vomerine tooth series width); 26) VTL (vomerine tooth series length); 27) UJTN (number of upper jaw teeth); 28) LJTN (number of lower jaw teeth); 29) VTN (number of vomerine teeth); 30) CGN (number of costal grooves following Misawa (1989)); and 31) LON (number of costal folds between adpressed limbs).

When the groups recognized in H. hirosei based on molecular analysis were compared (see Results), only adult males were used as H. hirosei is known to exhibit sexual dimorphism in most of morphological characters (Nishikawa et al., 2007) and a larger number of adult males than adult females was obtained. Sex and maturity were determined by direct gonad observations. To examine differentiation between the groups, we compared SVL using the Student’s t-test. The Mann-Whitney’s U-test was used to assess meristic characters and ratio values of metric characters to SVL. To examine the overall morphological variation between the groups, linear discriminate analysis (LDA) was conducted using loge-transformed metric values of the total of the 26 measurements. The difference in the LDA score between recognized groups was then determined using the Mann-Whitney’s U-test.

The clutch size among the populations was compared using Tukey’s pairwise post-hoc test.

In these analyses, the TAL data for the regenerated tail were omitted. All statistical analyses were performed using Past 4.04 (Hammer, Harper & Ryan, 2001).

For larval specimens, the developmental stage was determined according to Iwasawa & Yamashita (1991) and the SVL was measured accordingly.

Results

Mitochondrial DNA analysis

Complete sequences of the cyt b (1,141 bp) of mtDNA were obtained, which included 450 variables and 345 parsimonious informative sites. The best selected substitution models were the codon-equal rate model with the general time reversible model (GTR: Tavaré, 1986) + G for the first, second, and third in the ML analysis and the codon proportional model with SYM_Gamma (Zharkikh, 1994), HKY85_Gamma (Hasegawa, Kishino & Yano, 1985), and GTR_Gamma for the first, second, and third in the BI analysis, respectively. MCMC analysis using MrBayes did not converge; thus, only the phylogenetic tree constructed by ML (likelihood value [lnL] = −7,206.929; Fig. 1B) was presented. This tree indicated that a clade including H. amakusaensis, H. ikioi, H. osumiensis, and H. shinichisatoi, H. naevius, H. stejnegeri, the Tsurugi group in H. hirosei (BS = 100) from around the Tsurugi Mountains (localities 1–6, 10), H. oyamai, H. katoi, and a clade including H. sematonotos and the remaining samples of H. hirosei formed one clade (BS = 98), although the phylogenetic relationships within the clade were not well resolved. The remaining samples of H. hirosei were separated into the Central group (BS = 99) from the Sanuki Mountains (localities 8 and 9), the Takanawa Mountains (locality 19), and western part of the Shikoku Mountains (localities 10–18, 20–23); and a clade of the Nanyo group (BS = 100) from the Onigajo and Sasayama Mountains (localities 24–26). The Nanyo group had a sister relationship with H. sematonotos (BS = 99). Individuals of both the Tsurugi and Central groups were sympatrically found at locality 10.

The mean pairwise genetic distances (Table 2) were 9.4% between the Tsurugi and Central group, 10.3% between the Tsurugi and Nanyo group, and 6.9% between the Central and Nanyo group. The genetic distance between the Nanyo group and H. samatonotos was 4.6%.

Table 2 Uncorrected p-distances (in %; mean) for complete cyt b among the 16 hynobiid taxa compared.

		1	2	3	4	5	6	7	8	9	10	11	12	13	14	15	
1	Tsurugi group																
2	Central group	9.4															
3	Nanyo group	10.3	6.9														
4	H. sematonotos	9.8	6.2	4.6													
5	H. oyamai	9.8	9.3	10.0	9.6												
6	H. katoi	9.6	8.2	9.4	8.2	8.9											
7	H. naevius	10.8	10.8	11.3	11.3	11.2	11.2										
8	H. stejnegeri	10.9	11.0	11.8	11.2	10.3	10.6	11.7									
9	H. amakusaensis	11.8	12.8	12.9	12.8	12.8	11.8	12.5	12.2								
10	H. ikioi	12.1	12.7	13.0	12.9	12.4	11.8	12.2	12.4	2.9							
11	H. osumiensis	12.2	12.8	13.1	12.7	12.3	11.9	12.6	12.6	5.0	3.9						
12	H. shinichisatoi	11.7	12.4	12.6	12.1	11.9	11.5	11.9	12.4	8.1	7.8	8.1					
13	H. boulengeri	15.4	15.7	16.1	15.6	15.1	15.5	15.8	14.6	14.5	14.5	14.1	14.4				
14	H. kimurae	14.3	14.4	16.0	15.0	14.0	12.6	14.6	13.9	13.0	13.5	12.8	13.8	8.7			
15	H. retardatus	14.8	14.4	15.0	14.5	14.0	13.1	15.1	14.3	13.0	13.7	14.0	14.0	14.7	12.7		
16	S. keyserlingii	18.2	19.4	20.0	18.8	18.4	18.1	18.5	18.6	18.9	18.9	18.1	18.5	18.0	17.0	17.8	

SNP analysis

A total of 373,642 reads generated based on MIG-seq were filtered using fastp, resulting in 287,094 reads. Stacks detected 716 SNP loci in the reads, and the genotyping rate was 0.74. Based on STRUCTURE analysis, at K = 2, individuals were clustered into southeastern Shikoku (blue) vs. the remaining area of Shikoku (orange), and transitional structures were found at localities 8 and 9 (Sanuki Mountains) and 10–12 (Fig. 2). Locality 7 was included in the Central group by mitochondrial analysis, but was judged as the Tsurugi group in the STRUCTURE analysis. At K = 3, individuals from southeastern Shikoku (blue), the remaining area of Shikoku, except the Onigajo and Sasayama Mountains (orange), and the Onigajo and Sasayama Mountains (localities 24–26; green) were divided. These three groups recognized by mtDNA analysis were also detected using the SNP data; however, the Tsurugi and Central groups were not separated and clinally continued in the SNP data. Thus, we temporarily treated these two groups as one group for the morphological analyses.

Figure 2 Genetic structure of Hynobius hirosei sensu lato.

Top: haplotype of the mtDNA in each individuals. Middle and bottom: results of STRUCTURE analysis using SNP data.

Morphological analysis

To survey the morphological separation of the Nanyo group from the other groups, we conducted morphological analyses of the Nanyo and Tsurugi + Central groups. The obtained values of SVL and the ratios of the other characters to the SVL are shown in Table 3. The Nanyo group had significantly lower values for SVL (T = 2.8601, P = 0.0060), significantly larger values for RAGD (U = 80.00, P = 0.0002), RUEL (U = 146.00, P = 0.0156), and RMTAW (U = 167.00, P = 0.0448), and significantly lower values for RTAL (U = 123.00, P = 0.0061) and RIND (U = 130.50, P = 0.0065) than the remaining samples (Tsurugi + Central group).

Table 3 Means ± SD of SVL and medians of character ratios (R = % SVL), VTW/VTL, CG, LON, UJTN, LJTN, and VTN of the Nanyo group (Hynobius oni) and the remaining samples (H. hirosei). Ranges are shown in parentheses.

Species	Nanyo group (H. oni)	Tsurugi + Central group (H. hirosei)	
	Male	Female	Male	Female	
Characters	n = 12	n = 1	n = 45	n = 25	
SVL	79.4 ± 4.2	84.6	85.5 ± 7.1	94.244 ± 5.6	
	(73.6–87.5)		(73.0–102.2)	(82.3–104.6)	
RHL	23.4	24.0	24.1	23.0	
	(22.4–25.1)		(22.0–25.5)	(20.7–24.4)	
RHW	16.6	16.4	16.9	15.9	
	(15.8–17.6)		(14.4–18.7)	(14.0–16.8)	
RMXHW	17.7	18.0	18.2	16.2	
	(16.4–19.1)		(15.7–19.3)	(14.1–17.5)	
RSL	6.5	6.1	6.7	6.3	
	(6.1–7.1)		(5.5–7.6)	(5.8–7.1)	
RLJL	14.4	13.9	14.6	13.9	
	(13.5–15.2)		(12.9–16.2)	(13.0–14.7)	
RIND	6.0	5.9	6.2	6.1	
	(5.6–6.3)		(5.3–7.2)	(5.5–6.6)	
RIOD	5.3	5.0	5.3	5.0	
	(4.8–6.1)		(4.4–6.0)	(4.6–5.4)	
RUEW	3.9	3.3	3.9	3.8	
	(3.5–4.0)		(3.4–4.6)	(3.2–4.3)	
RUEL	6.0	5.8	5.6	5.4	
	(5.5–6.4)		(4.9–6.3)	(4.7–5.9)	
RAGD	54.0	52.5	52.1	54.4	
	(51.9–55.2)		(49.7–56.1)	(51.7–57.3)	
RTRL	76.6	76.0	75.9	77.0	
	(74.9–77.6)		(74.5–78.8)	(75.6–79.3)	
RTAL	80.4	77.9	84.2 (n = 43)	79.8 (n = 24)	
	(68.5–84.5)		(67.0–95.0)	(70.2–86.9)	
RBTAW	9.5	7.8	8.9	7.9	
	(8.7–10.2)		(7.4–10.5)	(6.6–10.4)	
RMTAW	7.1	5.1	6.2	5.4	
	(6.2–7.6)		(4.7–9.0)	(3.5–9.3)	
RBTAH	7.9	6.6	8.1	7.1	
	(7.2–9.6)		(5.2–10.6)	(5.0–9.4)	
RMXTAH	10.2	7.0	11.0	9.3	
	(8.4–12.0)		(8–15.2)	(6.2–12.4)	
RMTAH	9.9	6.3	9.7	8.3	
	(8.3–10.5)		(6.9–14.9)	(6.1–12.1)	
RFLL	24.5	23.0	23.9	22.4	
	(21.7–25.8)		(21.2–26.4)	(20–24.7)	
RHLL	31.1	30.9	30.2	29.2	
	(29.6–34.0)		(26.8–32.3)	(26.8–31.7)	
R2FL	4.5	4.7	4.7	4.1	
	(4.2–5.1)		(3.7–5.5)	(3.4–4.7)	
R3FL	4.5	4.1	4.7	4.4	
	(3.9–5.1)		(3.6–5.4)	(3.3–4.7)	
R3TL	7.1	7.0	7.0	6.6	
	(6.2–7.6)		(6.0–8.0)	(5.7–7.6)	
R5TL	3.0	3.3	3.2	3.0	
	(2.5–3.7)		(2.2–4.2)	(2.3–4.2)	
RVTW	6.2	6.1	6.3	5.9	
	(5.5–6.7)		(5–7.2)	(5.5–6.9)	
RVTL	4.1	3.9	4.0	3.9	
	(3.3–4.5)		(3.2–4.9)	(3.2–5.0)	
VTW/VTL	1.6	1.6	1.6	1.5	
	(1.3–2.0)		(1.2–2.2)	(1.2–1.8)	
CG	13.0	13	13.0	13.0	
	(12–13)		(12–14)	(13–14)	
LON	−1.3	−2	−1.5	−2.5	
	(−2 to 0)		(−2.5 to 0)	(−4 to −1)	
UJTN	76.0 (n = 11)	83	75.0	80.0	
	(65–87)		(66–91)	(66–93)	
LJTN	75.0 (n = 11)	84	72.0	76.0	
	(62–89)		(63–85)	(64–93)	
VTN	41.5	50	42.5 (n = 44)	47.0	
	(33–53)		(34–53)	(34–60)	

In the LDA results, the eigenvalues of the axis accounted for 2.79. On the axis, the highest absolute magnitude of the standardized discriminant coefficients was −266.35 of SVL, followed by that of TRL (224.66), HL (72.85), AGD (−27.06), and BTAW (−10.84). The Nanyo group and the remaining samples were significantly separated (U = 0.00, P = 0.0000), with a small overlap observed (Fig. 3).

Figure 3 Histogram of the first linear discriminant of the Nanyo group (Hynobius oni; green bar) and the remaining samples (H. hirosei; purple bar).

Systematics

In summary, the Nanyo group and the remaining samples were clearly separated based on the nuclear genome and external morphology, whereas the Tsusrugi and Central groups formed hybrid zones. The Nanyo group and the remaining samples were genetically isolated; thus, the Nanyo group must be an independent evolutionary lineage with sufficient genetic and morphological differentiation. This information strongly indicates that the Nanyo group is an independent species. The remaining samples (hereafter H. hirosei sensu stricto) included topotypic H. hirosei from Mt. Ishizuchi; however, the Nanyo group has not been named. Therefore, the Nanyo group was revealed to be a new species.

Hynobius oni sp. nov. urn:lsid:zoobank.org:act:D9C7A32A-7CB0-44F1-A0F2-202E7528EBEA

(Japanese name: Nan-yo-sanshouo)

(Figs. 4D–4F, 5–8)

Figure 4 Live male topotype of Hynobius hirosei (KUHE 62827; A–C) and male holotype of H. oni (KUHE 62785; D–F).

(A, D) Dorsal views. (B, E) Lateral views. (C, F) Ventral views. Scale bar shows 20 mm.

Figure 5 Live individual (KUHE 61096) of Hynobius oni.

Figure 6 Live larva of Hynobius oni.

(A) Dorsal view. (B) Left lateral view. (C) Ventral view.

Figure 7 Pair of egg sacs of Hynobius oni.

Scale bar = 10 mm.

Figure 8 Habitat of Hynobius oni in the type locality.

Pachypalaminus boulengeri: Sato (1934): 464 (part);

Hynobius (Pachypalaminus) boulengeri: Nakamura & Uéno (1963): 13 (part);

Hynobius boulengeri (population 9): Nishikawa et al. (2001): 281 (part).

Holotype: KUHE 62785, an adult male from Mt. Yatsuzura, Uwajima City, Ehime Prefecture, Shikoku, Japan (N 33.18°; E 132.62°; alt. 1,000 m a.s.l.), collected by S. Kanamori on April 24, 2021.

Paratypes: A total of 11 specimens: one female (KUHE 61097) from Mt. Onigajo, Uwajima City, Ehime Prefecture (N 33.19°; E 132.61°; alt. 980 m a.s.l.), collected by M. Hibino on May 3, 2019; one male (KUHE 61098) collected by Y. Tomimori on April 29, 2019, one male (KUHE 62327) collected by Y. Onishi on October 25, 2020, and eight males (KUHE 62784, 62786–62792) collected by S. Kanamori, K. Nishikawa, and Y. Suzuki on April 24, 2021, from the type locality.

Referred specimens: One male specimen (KUHE 61096) from Sozu, Ainan Town, Ehime Prefecture (N 33.04°; E 132.57°; alt. 730 m a.s.l.), collected by I. Fukuyama on May 2, 2019; and five juvenile specimens (KUHE 62320–62324) from the type locality, collected by Y. Onishi, S. Kanamori, Y. Tomimori, and I. Fukuyama on October 24, 2020.

Etymology: The specific name is derived from the “Oni” in Japanese, which is a traditional Japanese demon. The habitats of the new species are areas where prior generations believed that the Oni and Ushi-oni, which is a type of Oni, occurred. The type locality is located in the Oni-ga-jo Mountains, which is considered to be the castle of the Oni.

Diagnosis: A large-sized species (adult SVL 73.6–87.5 mm in males) of the lotic-breeding Hynobius, breeding in montane streams; dorsum uniformly dark reddish brown and immaculate in adult; tips of fore- and hindlimbs adpressed on body scarcely meeting (overlap of −2.0 to 0.0 costal folds in males); fifth toe well developed; ova large, pigmentless; egg sacs relatively long and crescent in shape, with distinct whiptail structure on free end; larvae lack claws on their tips of fingers and toes; most similar to H. hirosei, but distinct based on its smaller body size, longer axilla-groin distance, shorter tail length, shorter internarial distance, longer upper eyelid length, and larger medial tail width. Hynobius oni is genetically closer to H. sematonotos than H. hirosei based on mtDNA; however, H. oni has no large markings on the body, in contrast to many silvery spots on H. sematonotos, and a larger SVL than H. sematonotos.

Description of holotype: Head-body moderately large and robust; head oval and moderately depressed, distinctly longer than wide; snout rounded, slightly projecting beyond lower jaw; nostril close to snout tip; labial fold absent; eye large, prominently protruded, slightly inset from edge of head in dorsal view; upper eyelid well developed, shorter than snout; gular fold distinct, curving slightly anteriorly; parotoid gland evident, extending from angle of jaw to gular fold; postorbital grooves distinct, branching posterior to angle of jaw, one short and running down to lower jaw, the other long and posteriorly to parotoid gland; vomerine tooth series wider than long, V-shaped, anterior margin distal to choanae; tongue broad, both sides free from mouth floor; fore- and hindlimbs long and thick; number of costal grooves between axilla and groin 13; depressed limbs separated by two costal folds; relative length of fingers I<IV<III<II, toes I<V<II<IV<III; fifth toe well developed; cloaca longitudinal slit; genital tubercle on anterior cloaca absent; tail moderately short and thick, cylindrical at base, increasingly compressed posteriorly, dorsal fin evident posteriorly; tip of tail rounded in lateral view.

Measurements and counts of holotype (measurements in mm): SVL 82.7; HL 18.6; HW 13.6; MXHW 14.2; SL 5.4; LJL 11.2; IND 4.6; IOD 4.2; UEW 2.9; UEL 4.9; AGD 45.0; TRL 64.1; TAL 66.3; BTAW 8.4; MTAW 6.2; BTAH 6.5; MXTAH 9.3; MTAH 8.6; FLL 19.4; HLL 24.5; 2FL 3.7; 3FL 3.5; 3TL 5.9; 5TL 2.5; VTW 4.8; VTL 3.6; UJTN 71; LJTN 67; VTN 43.

Color: Dorsum uniformly dark reddish brown without marking; underside lighter than the dorsum; underside of tail slightly ochre; iris dark brown without marking. In preservatives, dorsal coloration tends to fade and becomes gray-brown, but otherwise has no obvious changes.

Variation: Morphometric data are presented in Table 3. Individuals of the type series and referred specimens are generally similar in body size and proportion. The single paratype female (SVL = 84.6 mm) is as large as the male holotype (82.7 mm). The female has smaller upper eyelid width (3.1%SVL vs. 3.2–3.8%SVL in males [n = 12]) and lower and thinner tail (BTAH: 7.8%SVL vs. 8.7–10.2%SVL; MTAH: 5.1%SVL vs. 6.2–7.6%SVL; BTAH: 6.6%SVL vs. 7.2–9.6%SVL; MXTAH: 7.0%SVL vs. 8.4–12.0%SVL; MTAH: 6.3%SVL vs. 8.3–10.5%SVL) than males (n = 12). The dorsal ground color is usually dark reddish brown, but is sometimes less reddish and darker. Young individuals have dorsum scattered by white dots.

Eggs and egg sacs: Hynobius oni has a smaller clutch size (mean: 23.5 ± 5.7, range: 16–36, n = 21) than H. hirosei sensu stricto from Mt. Ibuki, Ehime Prefecture (mean: 34.3 ± 7.6, range: 25–59; n = 81 (Tanabe & Okayama, 2004); df = 110, F = 16.01, P = 0.0000) and from the Tsurugi Mountains, Tokushima Prefecture (mean: 31.0 ± 12.4, range: 15–52, n = 9 (Tamura, 2012); df = 110, F = 16.01, P = 0.0356). Egg sacs are crescent in shape (length 176.8 ± 23.8 mm (n = 15), width 17.0 ± 1.9 mm (n = 15)) with relatively thicker envelope than that of other congeners, but thinner than that of H. boulengeri, and similar in thickness to that of H. naevius, H. oyamai, and H. sematonotos, with a distinct whiptail structure on the free end (length 35.2 ± 23.7 mm (n = 15)). Both the animal and vegetal poles of eggs have a cream color. The eggs (oval diameter: 4.9–5.9 mm) form a single and/or double row in each egg sac.

Larvae: SVL for fully grown larvae at stage 63 (n = 3) was 21.7 ± 1.5 mm, and SVL for a larva when onset of metamorphosis in late July at stage 64 (n = 1) was 20.6 mm; head rounded in dorsal and lateral views (Fig. 6); snout short and broadly rounded; eyes slightly protruded, inset from the edge of head in dorsal view; labial fold distinct at posterior half of upper jaw; external gills developed; caudal fin higher than head; dorsal fin higher than ventral fin; origin of dorsal fin at distal half to three-fourth of trunk; ventral fin originating from vent; tail tip weakly pointed; limbs slender; claws on fingers and toes absent. In life, dorsum light brown with small dark-brown dots and blotches; venter whitish and transparent; golden dots scattered on tail fin. In preservative, the dorsal coloration tends to fade, becoming light brown and golden dots fading to white.

Comparisons: Among species of the lotic-breeding Hynobius, H. oni with uniform dorsal color resembles H. boulengeri, H. katoi, H. shinichisatoi, H. osumiensis, and H. hirosei, whereas the remaining species in the group, H. kimurae, H. fossigenus, H. ikioi, H. amakusaensis, H. naevius, H. oyamai, H. sematonotos, H. stejnegeri, H. tsurugiensis, H. guttatus, and H. kuishiensis, have markings that can be easily distinguished from the new species. Hynobius oni is distinct from H. boulengeri (mean male SVL 79.4 mm vs. 93.9 mm in H. boulengeri) and H. hirosei (85.5 mm) based on its smaller body size, and from H. katoi (58.4 mm) and H. osumiensis (68.4 mm) based on its larger body size. Hynobius oni is distinguishable from H. shinichisatoi based on its shorter tail length (mean 80.4%SVL vs. 90.4%SVL in H. shinichisatoi) and shallower vomerine teeth series (VTW/VTL: 1.6 vs. 1.2 in H. shinichisatoi). In addition to the significantly smaller body size of H. oni than that of H. hirosei, H. oni is distinguished from H. hirosei using the following ratio values: longer axilla-groin distance, shorter tail length, shorter internarial distance, longer upper eyelid length, and larger medial tail width.

Range: Known only from the Onigajo and Sasayama Mountains in Ehime and Kochi prefectures, southwestern part of Shikoku Island, western Japan (Fig. 1A).

Natural history: Hynobius oni inhabits around mountain streams with partially exposed bedrock (Fig. 8). On April 29, 2019, egg sacs, some females with egg, and many males were observed in the water. On April 24, 2021, 12 adult males and egg sacs were found under stones in the water. Thus, the breeding season is presumed to be late April. Adults in the non-breeding season and juveniles were found under stones, rotten wood, and debris near the stream. However, the larval life history of H. oni is poorly understood. Only one overwintered larva (total length: 71 mm) was once found in an open stream through three-night surveys on April 29, May 3, 2019 and April 28, 2022. In the surveys, the observed density of overwintered larvae was lower (approximately 0.25 per 1 m2) than that of H. hirosei (more than 10 per 1 m2; our personal observation) on May 8, 2021. No sympatric hynobiid salamanders (Hynobius or Onychodactylus) were observed with H. oni.

Conservation: As the known range of the new species is restricted to small areas of only the Onigajo and Sasayama Mountains, habitat destruction and over-collecting for private purposes could have negative impacts on the natural populations, as reported for Onychodactylus tsukubaensis (Yoshikawa & Sakamoto, 2016), which also occurs in small areas (the Tsukuba Mountains of Ibaraki Prefecture). Hynobius oni has a very small range and some habitats have been degraded; thus, this species should be protected as an endangered species. Furthermore, the conservation status of H. hirosei should be reconsidered immediately after this taxonomic change (presently, listed as Near Threatened (Matsui, 2014)).

Discussion

Among the three groups of H. hirosei sensu lato observed in this study, the Nanyo group (Hynobius oni) was demonstrated to be an independent species based on genetic and morphological evidence. However, regarding the Tsurugi and Central groups, whether they are distinct species remains unclear until the nature of the hybrid zone is revealed, as they could not be genetically isolated from each other in the SNP analysis. Further analyses are necessary to determine the taxonomic status of the two groups by estimating the degree and demography of the hybrid zone, and elucidate the process of separation of the groups using robust DNA data, as reported by Burbrink & Ruane (2021).

Terrestrial animals inhabiting Shikoku tend to show genetically high inter- and intraspecific divergence (Kato, Morii & Tojo, 2013; Dejima & Sota, 2017; Tominaga et al., 2019b). The three groups with high divergence in H. hirosei sensu lato were distributed in different mountainous areas, and their distributions were usually separated by large rivers and lowlands. On the other hand, hybrid zones between the Tsurugi and Central groups were found, where mountains continued without being divided by large rivers. Large rivers are suggested to contribute to the distribution and maintenance of the population in H. hirosei sensu lato. Furthermore, the same pattern of distribution as the three groups in H. hirosei sensu lato was found in the harvestman, Pseudobiantes japonicus (Kumekawa et al., 2014, 2019), which is also a terrestrial and low-dispersal animal, similar to the salamander analyzed in this study. Therefore, the concordant pattern recognized in the salamander and harvestman suggests that large rivers may cause geographic variations in Shikoku.

Hynobius oni has a significantly smaller body size than H. hirosei sensu stricto. Based on our preliminary survey, H. oni tended to have a smaller number of overwintered larvae and possibly a smaller metamorphosing size than H. hirosei. In the Onigajo and Sasayama Mountains, where H. oni occurs, smaller numbers of stable streams with deep pools were found relative to other areas in Shikoku as the steep riverbeds were formed by exposed bedrock. Some streams dried up during winter. In such shallow and unstable water levels, larvae of lotic-breeding Hynobius do not overwinter often and do not grow to a large metamorphosing size (H. kimurae in Misawa & Matsui (1997); H. osumiensis in Nishikawa & Matsui (2008)), which agree with our preliminary field surveys in the Onigajo and Sasayama Mountains. Additionally, as shown in other congeners (Misawa & Matsui, 1997), a smaller metamorphosing size may induce smaller adult size in H. oni.

Terrestrial salamanders are well known to coexist through a partition of food resources based on their body size difference (Vignoli, Bissattini & Luiselli, 2016). In the Onigajo and Sasayama Mountains, only H. oni occurs. The smaller adult size of H. oni (79.6 mm in mean SVL) relative to that of H. hirosei sensu stricto could not permit its co-occurrence with other syntopic smaller-sized salamanders from Shikoku (e.g., H. kuishiensis with 60.8 mm, H. tsurugiensis with 62.0 mm (Tominaga et al., 2019b), and Onychodactylus kinneburi with 72.6 mm in mean SVL (Yoshikawa et al., 2013)).

In contrast, H. hirosei sensu stricto is a large species of the genus. Nishikawa et al. (2007) first reported that species of the H. boulengeri complex (including H. hirosei sensu stricto) tended to have a larger body size when they occurred sympatrically with other hynobiid salamander. Such interspecific morphological differentiation is known to occur to avoid interspecific competition for resources, such as food and habitat (Brown & Wilson, 1956; Adams & Rohlf, 2000; Melville, 2002). Indeed, H. hirosei sensu stricto occurs syntopically with H. kuishiensis, H. tsurugiensis, and O. kinneburi. Thus, this sympatric distribution might be possible due to the body size differences between them (Nishikawa et al., 2007).

The range of H. oni is one of the smallest among lotic breeding salamanders in Shikoku. Currently, this range has reduced owing to loss of habitat. Loss and fragmentation of habitats have the most negative impact on amphibian populations and their genetic diversity (Reh & Seitz, 1990; Frankman, Ballou & Briscoe, 2002; Cushman, 2006), especially species that have small distributions, such as H. oni. Presently, local populations of H. hirosei sensu lato are designated as CR+EN in the Red Data Book (RDB) of Matsuyama City (Okayama, 2012) and EN in the RDB of Kagawa Prefecture (Shinohara, 2021). Further, habitats in Kagawa Prefecture are limited and their population sizes are extremely small compared to those in other areas. The recent pet trade of rare salamanders have become a serious conservation problem (Terui & Tokuda, 2021). Therefore, legal protection of H. oni and H. hirosei sensu stricto by national and local governments is urgently needed.

Conclusions

In the present study, the actual genetic population structure and degree of genetic divergence within Hynobius hirosei, which has been reported to have large genetic intraspecies divergence, were evaluated using mtDNA and nuDNA markers (SNP). Phylogenetic analysis using mtDNA revealed three divergent lineages, including the Tsurugi, Central, and Nanyo groups (genetic distance: 6.9–10.3% in cyt b). Further, STRUCTURE analysis using SNP revealed that the Nanyo group is genetically isolated from the other groups and the Tsurugi and Central groups form hybrid zones. Morphological analyses also revealed that the Nanyo is distinct from the other groups. Collectively, these results strongly indicate that Nanyo group is a distinct species, and is referred to as H. oni sp. nov. in this study.

Supplemental Information

Supplemental Information 1 Raw measurements of Hynobius hirosei and H. oni..

Click here for additional data file.

Supplemental Information 2 Sequences of Hynobus hirosei and H. oni..

Click here for additional data file.

Supplemental Information 3 Mig-seq data used in STRUCTURE analysis.

Click here for additional data file.

We would like to thank T. Okayama, S. Ichihara, K. Araya, K. Eto, I. Fukuyama, R. Fukuyama, T. Hayashi, K. Hibino, H. Ishikawa, N. Koike, K. Kurita, N. Maeda, Y. Misawa, T. Nakano, G. Nakatsu, K. Niwa, Y. Onishi, T. Sota, T. Sugihara, Y. Suzuki, M. Tagawa, Y. Tomimori, Tsurugisan Chojo Hutte, M. Yoshida, E. Yamamoto, and N. Yoshikawa for their assistance with specimens collecting; Y. Tomimori for providing data related to his observations; K. Fukutani, I. Fukuyama, S. Hara, K. Kurita, and K. Niwa for assisting with the data analyses; Y. Fuke for his support regarding the experiment; and M. Munir for improving the earlier version of this manuscript.

Additional Information and Declarations

Competing Interests

Author Contributions

Animal Ethics

DNA Deposition

Data Availability

New Species Registration

The authors declare that they have no competing interests.

Sally Kanamori conceived and designed the experiments, performed the experiments, analyzed the data, prepared figures and/or tables, authored or reviewed drafts of the article, collected samples, and approved the final draft.

Kanto Nishikawa conceived and designed the experiments, performed the experiments, authored or reviewed drafts of the article, collected samples, and approved the final draft.

Masafumi Matsui conceived and designed the experiments, authored or reviewed drafts of the article, collected samples, and approved the final draft.

Shingo Tanabe conceived and designed the experiments, authored or reviewed drafts of the article, collected samples, and approved the final draft.

The following information was supplied relating to ethical approvals (i.e., approving body and any reference numbers):

Our research followed the Regulation on Animal Experimentation at Kyoto University (certificate number: 29–A–7, 30–A–7, 20–A–7, 20–A–5).

The following information was supplied regarding the deposition of DNA sequences:

MtDNA data are accessible via GenBank: ON110825–ON110940. MIG-seq data is available at DDBJ: PRJDB13424, BioSample: SAMD00467990–SAMD00468103, and DRA: DRX347415–DRX347528 (Experiment), DRR361514–DRR361627 (RUN).

The following information was supplied regarding data availability:

The MtDNA data is available at GenBank: ON110825–ON110940.

The MIG-seq data is available at DDBJ: PRJDB13424, BioSample: SAMD00467990–SAMD00468103, and DRA: DRX347415–DRX347528 (Experiment), DRR361514–DRR361627 (RUN).

The raw data are available in the Supplemental Files.

https://ddbj.nig.ac.jp/resource/bioproject/PRJDB13424

https://ddbj.nig.ac.jp/resource/biosample/SAMD00467990

https://ddbj.nig.ac.jp/resource/biosample/SAMD00468103

https://ddbj.nig.ac.jp/resource/sra-experiment/DRX347415

https://ddbj.nig.ac.jp/resource/sra-experiment/DRX347528

https://ddbj.nig.ac.jp/resource/sra-run/DRR361514

https://ddbj.nig.ac.jp/resource/sra-run/DRR361627

https://www.ncbi.nlm.nih.gov/nuccore/ON110825

https://www.ncbi.nlm.nih.gov/nuccore/ON110940.

The following information was supplied regarding the registration of a newly described species:

Publication LSID: urn:lsid:zoobank.org:pub:1EFFBF1B-1249-4A9F-8AB2-36BFCD6F6B9E

Hynobius oni sp. nov. LSID: urn:lsid:zoobank.org:act:D9C7A32A-7CB0-44F1-A0F2-202E7528EBEA

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
