# Peer review of "A new species of lotic breeding salamander (Amphibia, Caudata, Hynobiidae) from Shikoku, Japan"

_PeerJ, doi:10.7717/peerj.13891_

## Round 0.1 · original submission · Major Revisions

Dear authors, your contribution has been evaluated by experts in the discipline and their suggestions are enclosed. Additionally, I have observed that the abstract needs to be enriched by adding more details of methodology adopted and characteristics of new species. Moreover, authors need to highlight the pertinence of the research findings as last phrase of the abstract.

I strongly suggest adding more information in the introduction section pertaining to niche of salamander and significance of this study through effectively highlighting research and knowledge gaps.
Methodology section lacks citations at various places such as sample collection protocol etc.

Discussion lacks sufficiency and it is perhaps better enriching it by adding interpretation of recorded findings and adding more peer-findings.

Reviewer 1 ·

Basic reporting

The article reports quite interesting results which were obtained by using the latest tools and techniques of molecular biology however, there is minor lacking in the correct use of the English language. The manuscript needs revisions by authors (or their editing service provider or colleague) regarding correct use of articles, punctuations and the verbs wherever indicated in the annotated pdf file.
Introduction needs minor revisions.
Please revise the first sentences since these are confusing. Starting the introduction with an attractive and comprehensive way is recommended. Start first describing the genus or genera you want to introduce, its/their ecological distribution in the world, in Japan, and then move to the individuals species you are focusing here in this manuscript.

Experimental design

No Comments.

Validity of the findings

Please revise the Results Section as indicated by comments in the annotated file.

Additional comments

Discussion is lacking the debate on the possible predators of Hynobius hirosei and Hynobius oni. Moreover, discuss what habitats might be affecting the life of the species which might then led to such an evolutionary divergence to evolve into a newer species. Use some citations if available otherwise discuss based on your own knowledge based on observations.

Annotated reviews are not available for download in order to protect the identity of reviewers who chose to remain anonymous.

Reviewer 2 ·

Basic reporting

This manuscript is well written. I noticed a few confusing parts in the description of the results that need to be rewritten. Once those corrections are made, I will consider the manuscript acceptable for publication.

L177: In allometry of relative growth, the Log10-transformed values are generally known to show liner regression relationships between characters. So I think we usually the values transformed by Log10 and we had better to use the Log10-transformed values for multivariate analyses. I am not sure that Loge-transformed values also have liner regression relationships between character value.

L194: these tree >> this tree

L194-195: This sentence is not clear, please rewrite it. Is the first clade H. retadatus? I think we don’t regard only one sample as clade.

L195: Does the “one clade” in the end of this sentence [… formed one clade.] mean the group constituted by all Hynobius samples? But because you used only one outgroup taxa(Salamandrella keyserlingii), the monophyly of all Hynobius samples is not supported in this phylogeny. So, I think we cannot call all Hynobius samples as clade in the current result. If you want say this group as clade, you should used more than 2 samples of outgroup samples.

L195: “the third clade”. If you regarded “H. retadatus” as the first clade, this is not adequate, so “the third clade” also should be reworded.

L207-208: This sentence is not clear. I suggest the sentence should be reworded as below. “The genetic distance between the Nanjyo group and H. samatonotos was 4.6%.” or something like that.

L213, 229: “Nanjyo group” or “the Nanjyo group”? I think it should be consistent throughout this manuscript.“Tsurugi group” and “Central group” also should be consistent in this manuscript.

L214: “connected” should be replaced to “merged”, “hybridized”, or “mixed”

L345: smaller >> lower

Experimental design

The experimental design of this study is quite appropriate. There may be some analyses that could be reconsidered.

Validity of the findings

I think that the findings of this study will contribute to our understanding of the regional amphibian fauna and are important for the evolutionary history of this group in Japan.

Reviewer 3 ·

Basic reporting

This paper aims to reveal geographic variations and to re-evaluate taxonomic status of a hynobiid salamander species, Hynobius hirosei using genetic and morphological approaches, and describing a new species. It seems to be well written in clear, unambiguous English and structured to meet the journal's standards. The Introduction section well review background of the target species referring related previous studies. There is no waste in figures and these are correctly captioned, although I think there is deficiency in Figure 2 (Structure barplot).
Raw data supplied seemed to be no problem although I could not check the registered sequence data using accession numbers described in the paper (these data appear not to be available yet?). Measurements in supplementary table may include some errors to be corrected (see general comments).
Nomenclatural acts made in this paper is in accordance with the International Commission on Zoological Nomenclature.

Experimental design

This is original and latest research in a series of systematic study of a group of lotic breeding salamander. Research question of this paper is well defined and meaningful (to reveal geographic variation of H. hirosei and to solve taxonomic problem by describing a local group as a new species). Methodology used in this study is well documented and is sufficient for scientific reexamination.

Validity of the findings

Result of this paper is based on robust data of genetic (mtDNA and nuDNA) and morphology in general, and no major problems with statistical analysis. Conclusion is also based on the data presented and well documented.
This paper presents numerous data of satisfied quality in Results section, however, Discussion section is not enough I think. Because it is too concentrated on the new species described (H. oni) and completely lacking mention on phylogenetic relationships of H. hirosei sensu lato and other related species, overall genetic structure of H. hirosei sensu lato, and new insights on taxonomic status of Central and Tsurugi groups. Especially, it is unclear how does the authors think about taxonomic status of Tsurugi group, which is also genetically divergent in both of the genetic markers. Current situation of hybrid zone between the Tsurugi and Central groups does not also appear in Discussion. These parts are better to be improved by adding discussion on the issues listed above.

Additional comments

My specific comments are listed below.

Line 126
“genomic” --> “genetic”
Genomic structure means arrangements of genes or other elements in genomic sequences or chromosomes, I think.

Line 202-204 “…which locates near the boundary between Tsurugi and Central groups.”
It is natural because the boundary is defined as the locality where Tsurugi and Central groups occur sympatrically. This part is unnecessary.


Line 212-214
You should describe result of STRUCTURE analysis by genomic SNPs independent from the separation of mtDNA result at first. Recognition of three group is not a premise.
In addition, I strongly recommend to add barplot at K=2 in Fig. 2, to show order of separation of genetic clusters.

Line 215-216
“localities 7 and 8” may be erroneous. “localities 8 and 9” may be correct.
No transitional structure is seen in localities 12-14. Do you mean 10-12?

Line 216-218 “The locality 9 was included in Central group by mitochondrial analysis, but all individuals of locality 9 were judged as Tsurugi group in the result of STRUCTURE analysis (Fig. 2).”
Do you mean locality 7? Unless so, it does not make sense.

Line 218-219
I am not sure this interpretation is plausible or not because you only present STRUCTURE barplot at K=3. It only indicates H. hirosei sensu lato is divided into three groups, not two major groups, and two of them are in contact to form hybrid zone. I think you should additionally present barplot at K=2 to show separation of Nanyo group and others is evident.

Line 254 “A total of 12 specimens”
It is 11 I think.

Line 274 “Hynobius oni is genetically closer to H. sematonotos than H. hirosei, ” --> “Hynobius oni is genetically closer to H. sematonotos than H. hirosei in mtDNA, “
You should clearly mention that close relationship is observed in mtDNA, because phylogenetic relationships in genomic DNA is unknown.

Line 311 “with slightly thin envelope”
Slightly thin compared with what? All of lotic breeding Hynobius have relatively thick eggsac envelope than that of lentic breeding Hynobius. I think envelope of H. oni is also thicker, not thin like lentic Hynobius.

Line 326
“kuishiesis” --> “kuishiensis”

Line 377-386
What mentioned in this paragraph is similar to that mentioned in the first paragraph in this section. This part can be combined and shortened.

Line 391-392 “which made the range of H. hirosei sensu stricto become smaller.”
How much became smaller by separation of H. oni?

Legend of Figure 4
“…KUHE 62785; G-I) in life.” --> “…(KUHE 62785; D-F) in life.”

Legend of Figure
Is this specimen not included in type materials or referred specimens? Why you did not used a photo of type specimens?

Caption od Table 2
Are these values averages? If so, mention it in the caption.

Table S1
Names of morphological characters are with ”R”, which means value relative to SVL in this study. It is not correct.
In addition, measurement of UEW of the holotype of H. oni (KUHE62785, 2.9mm) is different from the main text (line 291, 2.8mm). Please check which is correct.

---

## Round 0.2 · accepted · Accept

Authors have carefully incorporated suggestions forwarded by reviewers and editorial comments.

Reviewer 2 ·

Basic reporting

The revised manuscript is well written with adequate kinds of literature. The contents will attract many researchers, especially taxonomists of amphibians. I have no more additional comments.

Experimental design

I think all of the experimental designs are appropriate and essential to the authors' conclusions.

Validity of the findings

The new species revealed by the authors is an important finding for the biogeography of the region and for the study of the evolutionary history of the genus Hynobius.

Reviewer 3 ·

Basic reporting

no comment

Experimental design

no comment

Validity of the findings

no comment

Additional comments

I checked the MS and confirmed that it is revised and enriched well based on the comments raised by reviewers and the editor. I think This MS is worthy to be accepted for publication in PeerJ.